# Integrated Multitrophic Aquaculture; Analysing Contributions of Different Biological Compartments to Nutrient Removal in a Duckweed-Based Water Remediation System

**DOI:** 10.3390/plants11223103

**Published:** 2022-11-15

**Authors:** Simona Paolacci, Vlastimil Stejskal, Damien Toner, Marcel A. K. Jansen

**Affiliations:** 1School of Biological, Earth and Environmental Sciences, University College Cork, Distillery Fields, North Mall, T23 N73K Cork, Ireland; 2Faculty of Fisheries and Protection of Waters, South Bohemian Research Center of Aquaculture and Biodiversity of Hydrocenoses, Institute of Aquaculture, University of South Bohemia in Ceske Budejovice, Husova Tř. 458/102, 370 05 České Budějovice, Czech Republic; 3BIM, Ireland’s Seafood Development Agency, Crofton Rd, Dun Laoghaire, A96 E5A0 Co. Dublin, Ireland; 4Environmental Research Institute, Lee Road, T23 N73K Cork, Ireland

**Keywords:** Lemnaceae, aquaculture effluents, IMTA, RAS, phytoplankton, bacteria, phytoremediation

## Abstract

Duckweed (Lemnaceae) can support the development of freshwater aquaculture if used as extractive species in Integrated MultiTrophic Aquaculture (IMTA) systems. These aquatic plants have the advantage of producing protein-rich biomass that has several potential uses. On the contrary, other biological compartments, such as microalgae and bacteria, present in the water and competing with duckweed for light and nutrients cannot be harvested easily from the water. Moreover, as phytoplankton cannot easily be harvested, nutrients are eventually re-released; hence, this compartment does not contribute to the overall water remediation process. In the present study, a mesocosm experiment was designed to quantify the portion of nutrients effectively removed by duckweed in a duckweed-based aquaculture wastewater remediation system. Three tanks were buried next to a pilot-scale IMTA system used for the production of rainbow trout and perch. The tanks received aquaculture effluents from the adjacent system, and 50% of their surface was covered by duckweed. Daily water analyses of samples at the inlet and outlet of the mesocosm allowed quantification of the amount of nutrients removed in total. The portion removed by duckweed was determined by examining the nutrient content in the initial and final biomass. The portion of nutrients removed by other compartments was similarly estimated. The results show that duckweed is responsible for the removal of 31% and 29% of N and P, respectively. Phytoplankton removed 33% and 38% of N and P, respectively, while the biofilm played no major role in nutrient removal. The remainder of the removed nutrients were probably assimilated by bacteria or sedimented. It is speculated that a higher initial duckweed density can limit phytoplankton growth and, therefore, increase the portion of nutrients removed by the duckweed compartment.

## 1. Introduction

Demand for protein is rising sharply, with worldwide shortages of quality protein expected in the nearby future [1]. Aquaculture has the potential to contribute substantially to the production of protein required to feed an increasing world population [2]. Freshwater aquaculture can be a local source of protein, even in regions distant from the coast, where supply of marine seafood would involve food miles, and associated carbon emissions [3]. 

The development of freshwater aquaculture is sometimes impeded by concerns of negative environmental impacts. Consequently, the development of innovative, sustainable approaches to aquaculture is increasingly recognized as central to accelerate growth of the sector [4]. Such sustainable aquaculture should focus on high yields of quality produce, as well as minimise negative effects on the environment. Two negative effects of traditional freshwater aquaculture relate to water-use and eutrophication. Intensive aquaculture generates effluents rich in dissolved inorganic nutrients such as ammonia, and phosphate. If discharged without treatment, these effluents can have strong negative impacts on the water quality of receiving waterbodies. Eutrophication can lead to excessive growth of phytoplankton leading to algal blooms may, in turn, cause hypoxia which can affect a broad spectrum of organisms ranging from invertebrates to fish [5]. Moreover, the use of large volumes of freshwater for traditional aquaculture can exert pressure on local water resources in drier regions [6].

Recirculating Aquaculture Systems (RAS) can reduce the amount of water necessary to farm fish and prevent the negative effects of nutrients released by aquaculture on aquatic ecosystems [7]. In such recirculating systems, water is partially reused after undergoing remediation treatment using algae [8], bacteria [9] or aquatic plants [10]. Recently, several papers have described the use of duckweed as part of RAS system [11,12,13]. The use of duckweed in RAS has a promising future as it addresses two separate problems simultaneously; (1) it reduces the impact of aquaculture by capturing plant nutrients and (2) it converts a waste product into a resource that has an economic value [11].

The term duckweed refers to a group of freshwater free-floating plants belonging to the family of Lemnaceae. These plants are characterized by high growth rate and high protein content [14]. Thanks to their opportunistic nature, these plants thrive in eutrophic environments [15] and their use for the treatment of wastewater has been amply demonstrated, with early papers going back as far as 1973 [16]. In recent years, there has been a renewed interest in these species and their use for the treatment of a range of different agri-food wastewaters [17,18,19]. Rapid growth is associated with a high capacity to extract nutrients from water. Rapid growth similarly results in a high capacity to generate valuable biomass. Due to their high protein content, the use of these plants has been suggested as an alternative source of protein for livestock [20]. Alternatively, these plants can substitute conventional synthetic fertilizers [21] or be used as a biofuel [22].

A duckweed-based remediation system is characterized by a dynamic balance between the main compartments: duckweed, phytoplankton, biofilm, and other photosynthetic bacteria. Each group forms a functional compartment that affects the other two, and the co-existence of the three compartments determines the remediation efficiency of the whole system. The balance between duckweeds, algae and bacteria changes seasonally [12]. A reduced mat of duckweed in winter and spring is associated with a relative increase in phytoplankton. In contrast, during the summer, rapid growth of duckweed results in the formation of a conspicuous duckweed mat on the water surface, resulting in the shading of the underlying water column [12]. This, in turn, will impede algal photosynthesis, and growth, and ultimately the sequestration of nutrients from the water by algae [13,23]. Understanding of the relative proportion of nutrients removed by different taxa is important in order to make accurate models on water quality, and to inform IMTA management. Moreover, the removal of nutrients by taxa other than duckweed, will affect the portion of nutrients effectively recovered from the aquaculture system, as only duckweed can be readily harvested. In turn, this means that the competition for nutrients will also determine the yield of valuable biomass, and therefore the commercial outcome from a duckweed-based IMTA [23].

In the present study, an experiment was designed to quantify the relative uptake rate of nitrogen and phosphorus by duckweed species, algae and bacteria, in a duckweed-based aquaculture wastewater restoration system. The experiment aims to improve understanding of the balance between the different biological compartments in order to develop best practices for the management of these systems. The knowledge produced will help to optimize water restoration and maximize biomass production.

## 2. Materials and Methods

The experiment was carried out at an Integrated Multitrophic Aquaculture (IMTA) fish farm (Co. Offaly Ireland, coordinates 53.275555, −7.208392) where Oncorhynchus mykiss and Perca fluviatilis are farmed. The aquaculture effluents produced in four fish ponds are sent to a system of canals where the duckweed species *Lemna gibba* and *Lemna minor* are used to remove plant nutrients from the water. After treatment by duckweed, the water is returned to the fishponds. The technical details of this IMTA system have been extensively described by [13]. The experiment was carried out in August 2020, when suitable conditions for duckweed growth were present, and the effluent treatment canals were abundantly covered in *L. gibba* and, to a lesser extent, *L. minor*.

### 2.1. Experimental Set-Up

A small model of a duckweed-based effluent treatment system was set up next to the full-scale system. This mesocosm consisted of three independent tanks (i.e., three replicates) into which fresh aquaculture effluent was pumped. 

Prior to the experiment, six tanks (area 35 × 60 cm, depth 45 cm) were submerged in one of the duckweed-covered canals for seven days (Figure 1) to establish a biofilm of microorganisms and sediment on the internal surface of the tanks. Out of these six tanks, three tanks were subsequently used as part of the experimental set-up, while a further three tanks were used to analyze the biofilm present at the start of the experiment. After having been submerged for seven days in a canal, the latter three tanks were air-dried for three days, and the dry biofilm was used to determine the organic fraction and Total Nitrogen (TN) and Total Phosphorous (TP) contained within the biofilm at the start of the experiment.

The experimental tanks were buried next to one of the IMTA treatment canals. Just 5 cm of the side of the tank was left above ground level. At the start of the experiment, the tanks were filled with 70 L of aquaculture effluent, and 40% of their surface area was covered with duckweed, both taken from the neighboring IMTA system. An equal amount of duckweed biomass was collected, weighed, dried at 60 °C for three days, and analyzed for TP and TN. These values of TP and TN represent the initial P and N content in duckweed biomass.

A small pump was placed inside each tank in order to generate internal effluent recirculation. The flow rate was 490 L·h^−1^, a value that is similar to the flow conditions in the IMTA canals (Figure 2.1). Every 12 h, a sample of water was collected from each replicate mesocosm tank, after which the tanks were gently drained without disruption of the biofilm or the duckweed biomass (Figure 2.2). The water removed was immediately replaced with fresh aquaculture effluent, pumped in from the adjoining canal (Figure 2.3). A sample of effluent pumped in from the canal was also collected for analysis in order to quantify the initial nutrient concentration in the water before being treated. The collected water samples were filtered to separate phytoplankton from the remainder of the water sample. TP and TN were quantified both in filtered water and in phytoplankton. Furthermore, algal chlorophyll content, cyanobacteria, and turbidity (expressed in Formazin Turbidity Unit, FTU) were measured both in the tanks and in the canal using an AlgaeTorch produced by Bbe-Modanke. A Seneye online system (Seneye Ltd., Norwich, UK) is present in the system to monitor pH and temperature.

The mesocosm experiment was terminated when the duckweed achieved a tank surface coverage close to 90%; this happened after eight days. At this stage, the duckweed biomass was harvested, dried at 60 °C for 3 days, and analyzed for TP and TN. The difference in the total amount of nutrients in the final duckweed biomass versus the initial duckweed biomass (i.e., plant concentration times biomass at start and finish) represents the TN or TP removed by duckweed.

Once the duckweed was removed, the tanks were emptied and air-dried for three days. The internal biofilm was gently removed and weighed. Half the biofilm biomass was used for the quantification of TP and TN, while the other half was retained to determine the organic and inorganic components. The difference between the total nutrient content in the final biofilm versus the initial biofilm (i.e., concentration times biomass at start and finish) represents the TN or TP removed from the water by bacteria, periphytic and epipelic algae, and other biofilm constituents. The experimental protocol is summarised in Appendix A.

Half of the volume of water sampled was analyzed without filtering it, while the other half was filtered using a vacuum pump and a 5 µm filter. The phytoplankton biomass present in each sample was assessed by weighing the clean filter and then weighing it again after water sample filtration.

In the present paper, the term “duckweed” refers to the plant and its microbiome, and the term “biofilm” is used to indicate the mix of microorganisms (algae, protista, bacteria, and otherwise) and inorganic particles that build up on the internal surface of the tanks and contribute to nutrients sequestration. The term “phytoplankton” is used to indicate all the organisms with a diameter greater than 5 µm that live in the water column. Samples of biofilm and phytoplankton were observed under a Leica DM500 light microscope, and the microalgae were identified at the genus level using the dichotomic key presented in [24].

### 2.2. Analytical Methodology

TN in the unfiltered water sample was determined using an automated colorimetric method involving digestion of the unfiltered sample with potassium persulphate and boric acid in an alkaline solution in an autoclave at 121 °C for 30 min [25]. TP in the unfiltered water was determined using a modified Molybdate—Ascorbic Acid method following digestion of the unfiltered sample with persulphate and sulphuric acid in an autoclave at 121 °C for 30 min [26]. 

The other half of the samples were filtered and the dry weight of phytoplankton (mg·L^−1^) was determined. Total Dissolved Nitrogen (TDN) and Total Dissolved Phosphorus (TDP) were determined in the filtered effluent sample using the same method as detailed above. The difference between TN and TDN was taken as the total nitrogen present in phytoplankton. TP in phytoplankton was determined using the same approach. The amount of nutrients removed by phytoplankton was estimated by calculating the difference between TN and TP in the phytoplankton contained in the inlet and outlet water.

Plant samples, dried at 60 °C for 48 h and milled, were digested with concentrated sulphuric acid and Kjeltab Cu/3.5 in a TECATOR 2040 Digestor at 420 °C for 1 h. Digested samples were diluted to 250 mL using deionised water. The Total Kjeldahl Nitrogen was analysed using QuickChem IC + FIA flow injection analyzer (8000 series) manufactured by Lachat Instrument. For the determination of TP, the samples were acid digested and analysed using the ammonium molybdate method (Murphy and Riley, 1962). Absorbance was measured using a UV-Visible Recording Spectrophotometer manufactured by SHIMADZU Corporation (Model: SHIMADZU UV-160A). The same procedures were used for the analyses of TN and TP in the biofilm.

### 2.3. Duckweed Growth Analysis

A photograph of the duckweed mat was taken every morning and analysed using the imaging software ImageJ to determine the growth in the duckweed covered tank surface area. The Relative Growth Rate (RGR) was calculated using the formula by [27]:RGR = ln(Yf/Yi)/t
where Yi is the initial area of duckweed cover, Yf is the final area, t is the time in days and ln is the natural logarithm.

### 2.4. Calculations

The total amount of nitrogen removed from the water during the 8 days of experiments is expressed with the formula:TNr = TNd + TNb + TNp + TNo
where:

TNr = Total Nitrogen removed from the wastewater

TNd = Total Nitrogen removed by duckweed

TNb = Total Nitrogen removed by the biofilm

TNp = Nitrogen removed by phytoplankton

TNo =Nitrogen removed by other processes/organisms

The variables of this equation were calculated as follows:$$\mathrm{TNr}\;=\sum_{day\;1}^{day\;8}\;\left({\mathrm{CN}}_{i}\times \;V\right)-\left({\mathrm{CN}}_{o}\times \;V\right)$$

CN_i_ = concentration of nitrogen in the inflow (filtered water pumped in from the canal) expressed in mg·L^−1^

CN_o_ = concentration of nitrogen in the outflow (water in the tank after 12 h) expressed in mg·L^−1^

V = volume of water treated in 12 h expressed in l
TNd = (CNd_f_ × dwd_f_) − (CNd_i_ × dwd_i_)

CNd_f_ = concentration nitrogen in final duckweed biomass 

wd_f_ = final duckweed weight

CNd_i_ concentration of nitrogen in initial duckweed biomass

wd_i_ = initial duckweed weight
TNb = (CNb_f_ × dwb_f_) − (CNb_i_ × dwb_i_)

CNb_f_ = concentration of nitrogen in the final biofilm

dwb_f_ = final biofilm dry weight

CNb_i_ = concentration of nitrogen in the initial biofilm

Dwb_i_ = initial biofilm weight
$$\mathrm{TNp}\;=\sum_{day\;1}^{day\;8}\;\left({\mathrm{CNp}}_{f}\times {\mathrm{wp}}_{f}\right)-\left({\mathrm{CNp}}_{i}\times {\mathrm{wp}}_{i}\right)$$

CNp_f_ = concentration of nitrogen in phytoplankton in the outflowing water

wp_f_ = final phytoplankton weight

CNp_i_ = concentration of nitrogen in phytoplankton in the inflowing effluent

wp_i_ = initial phytoplankton weight

After determining the variables described above, TNo was determined as a difference between the TNr and TN extracted by the other compartments:TNo = TNr − (TNd + TNp + TNb)

The total amount of phosphorus removed from the water during the eight days of experiments was expressed with the same formula described above for the total amount of nitrogen removed.

### 2.5. Statistical Analyses

R studio version 4.1.1. was used for the statistical analysis. The correlation between the area of the duckweed mat and N and P removed from the tanks was analyzed with the non-parametric Kendall Tau test. A t-test was used to analyze differences in nutrient removal between night and day. 

## 3. Results

### 3.1. Water Parameters

During the experiment, the water temperature varied between 11 and 19 °C.

The pH of the aquaculture effluent varied between 7.55 and 8.53 during the 8-day experiment, while the turbidity varied between 20.9 and 24.6 FTU (Figure 3).

The total dissolved nitrogen and phosphorus concentrations in the incoming aquaculture effluent were quantified twice a day, i.e., every time the water in the tanks was replaced. TN varied between 2.97 and 4.3 mg·L^−1^, while TP varied between 0.43 and 0.72 mg·L^−1^ (Figure 4).

The concentration of algal chlorophyll and cyanobacteria in the incoming effluent was also measured twice a day. The chlorophyll concentration in the water varied between 251.5 and 342.2 µg·L^−1^. The concentration of cyanobacteria was extremely high on the first day of the experiment (66.1 µg·L^−1^), but it decreased during the following days, reaching a low of 21.1 µg·L^−1^ toward the final days of the experiment (Figure 5).

Samples of biofilm attached to the internal surface of the tanks were observed with a microscope. The genera identified are indicated in Table 1.

### 3.2. Nutrient Removal

The total volume of effluent treated by each tank in eight days was 1050 L. This volume, together with the cumulative differences in nutrient concentration between inflow and outflow water, allowed the calculation of the amount of dissolved N and P removed from the water.

A total of 689.93 mg of nitrogen (Figure 6) and 154.44 mg of phosphorus (Figure 7) were removed from the aquaculture wastewater. These numbers were estimated from the difference between inflow and outflow water in unfiltered samples. The N removal rate varied between −0.16 and 2.63 mg·L^−1^·d^−1^, while the P removal rate varied between 0.05 and 0.64 mg·L^−1^·d^−1^. There was no significant difference between N and P removed during the day and during the night.

### 3.3. Nutrients Accumulated by Phytoplankton

The phytoplankton contained in the inflow and outflow water was filtered and weighed. The filtered water samples (not containing phytoplankton) were analyzed for N and P content. The difference in nutrient concentration between filtered and unfiltered water samples represents the net nutrient removal by this biological compartment. The phytoplankton captured through filtering of the inlet and outlet water allowed the estimation that an average of 32.64 g of fresh weight phytoplankton (13.88 ± 1.22 during night hours and 18.66 ± 2.46 g during day hours) were produced, in each tank during the eight days of the experiment. The TN and TP concentrations in this biomass ranged between 14.82 and 35.2 mg N·g^−1^ and 5.72 and 10.09 mg P·g^−1^. The sums of TN and TP accumulated by phytoplankton are in total 230.06 mg (Figure 8) and 59.08 mg, respectively (Figure 9). Table 2 shows the details of phytoplankton produced and nutrients accumulated by this compartment.

### 3.4. Duckweed Biomass

The area of the duckweed mat increased from 831.81 ± 23.03 cm^2^ to 1936.47 ± 455.54 cm^2^. The RGR calculated for the duration of the experiment was 0.11 ± 0.003 d^−1^. Figure 10 shows the daily increase in duckweed surface. The plants grew slowly during the first days, while the absolute growth rate increased over the last two days.

The relationship between the increase in the area of the duckweed mat and the removal of nutrients was not linear (Appendix A); the non-parametric Kendall Tau test failed to identify a significant correlation between the duckweed area and the amount of N and P removed from the water.

The dry weight of duckweed, during the experiment, increased by 5.81 g. The content of N and P in duckweed biomass is indicated in Table 3. The aquatic plants removed, in 8 days, 210.1 mg of N and 44.5 mg of P from the entire water volume treated.

### 3.5. Biofilm

Most of the biofilm on the internal surface of the tanks constituted inorganic sediments. The organic matter represented only 15.6% of the biofilm at the start of the experiment and only 3.4% at the end of the experiment. The biofilm did not contribute substantially to the removal of nutrients from the wastewater. On the contrary, 11.2 mg of N and 4.9 mg of P were released from this compartment over the eight-day experimental period (Table 4).

### 3.6. Nutrient Removal from Different Compartments

The calculations illustrated in Section 2.5. were used to estimate the relative nutrient uptake by the different biological compartments. Duckweed is responsible for the uptake from the wastewater of 31% of the total nitrogen removed, while phytoplankton is responsible for another 33% of nitrogen removal. Some 29% of total phosphorus removal can be attributed to duckweed, while 38% is removed by phytoplankton. The biofilm did not contribute to the removal of nutrients. The remaining 36% and 33% of N and P, respectively, were removed from the aquaculture effluent by other means, such as denitrification and volatilization (Figure 11).

## 4. Discussion 

The present study assessed the amount of N and P removed from aquaculture wastewater by duckweed species, biofilm, and phytoplankton. The respective proportion of nutrients removed by these three compartments will inform system management and predict the portion of the nutrient load that can effectively be converted into valuable biomass as part of a circular economy approach.

### 4.1. Characteristics of the Effluent

The water quality parameters observed during the experiment are consistent with the parameters observed the year before by O’Neill et al. [28] during the same season, and at the same aquaculture farm. The nutrient concentration observed are also in the range previously identified by Paolacci et al. [29] The latter authors reviewed the characteristics of wastewater generated by rainbow trout and perch farms. They found that TN ranges between 0.5 and 70 mg·L^−1^, while TP ranges between 0.42 and 15 mg·L^−1^, depending mainly on the fish density.

The chlorophyll concentration is an indicator of the phytoplankton density in the water. Phytoplankton plays an important role in recirculating systems as it removes the NH4+-N from the effluents, contributing to the water restoration process [12]. The average chlorophyll concentration measured by [30] in a rainbow trout farm was 5 μg·L^−1^, considerably lower than the concentrations observed in the effluents used for this experiment (between 251.5 and 342.2 µg·L^−1^). However, Sen and Sonmez’ data [30] refer to a flow through system. In an outdoor recirculating system the phytoplankton is never released into natural waters and it seems plausible that higher chlorophyll concentrations can accumulate in the water.

Water turbidity depends mainly on the concentration and the characteristics of the suspended particles which depend, amongst others, on the geological substrate [31]. The experimental site is located in a cutaway peatland and the values of turbidity measured are consistent with those measured by [32] in lakes present in the same area (Offaly, IE), and generated by flooding cutaway bogs without removing the residual peat (as it was done in the IMTA system of the present study). Turbidity affects light diffraction in the water column and can limit phytoplankton growth. Despite the high turbidity, the chlorophyll concentration measured suggests that this parameter was not suppressing phytoplankton. However, the turbidity can explain the low number of taxa observed in the biofilm and in the phytoplankton community [33].

### 4.2. Nutrient Removal from Water

The results show that the total amount of N removed from the water was nearly five times higher than the amount of P removed (684.9 g vs. 154.4 g). This proportion is consistent with the observations of other authors. For example, [34] reported that, in a duckweed-covered sewage lagoon, the removal rates for N and P were, respectively, 0.26 g·m^−2^·d^−1^ and 0.05 g·m^−2^·d^−1^. This proportion reflects the N:P ratio in duckweed biomass [34] and also the N:P ratio observed in the biomass of other aquatic plants [35]. Similar N and P uptake data were also previously reported by [36] using dairy processing waste.

No significant difference was observed in nutrient removal during the night and the day. This can be explained by considering that the samples were taken at 8 am and 8 pm, in the summer, in Ireland. Between July and August (when the experiment was performed) this latitude experiences around 16 h of daylight. This means the ‘night’ uptake includes four hours of light during which duckweed and phytoplankton can perform photosynthesis and remove nutrients. Moreover, during night hours, sedimentation of nutrients is active and can contribute to N and P removal from the water column.

During the experiment, duckweed increased their surface density from 50% to >90%. The growth rate was consistent with the growth rates observed earlier for duckweed during the same time of the year in Ireland [37]. Around 30% of both N and P in the water was removed by duckweed, presumably to support growth, although some luxury uptake cannot be excluded. This percentage is consistent with the results of a small-scale experiment (1 L batches) performed by [38]. The latter authors determined the uptake rates of N and P in domestic wastewater by duckweed, algae and bacteria, at different initial densities of duckweed. They observed that duckweed was responsible for removing between 30% and 47% of N and up to 54% of P, depending on the initial plant density. The results of the present study are also in accordance with the observations of [23] who performed a competition experiment between phytoplankton and duckweed. When the duckweed mat was dense (like in the present study), the authors observed an increase in N and P content in the plant biomass, while at lower duckweed density the phytoplankton was competing with duckweed for nutrients, and this resulted in a decreased content of N and P in the biomass. The values reported by [23] for N and P content at high duckweed density are 36.65 and 10.95 mg·g^−1^ respectively, while the values observed in the present study are 36.8 (N) and 7.99 (P) mg·g^−1^ of biomass. The slightly lower value for P content is probably due to a lower concentration of this element in effluents used in the present study.

It is important to highlight that the removal of nutrients by duckweed measured in this study was determined by analysing N and P content in the initial and in the final biomass harvested at the end of the experiment. As a consequence, the values include the indirect contribution of algae and bacteria attached to, or incorporated into, the duckweed.

The phytoplankton removed on average 30 μg·L^−1^·d^−1^ of N and 7 μg·L^−1^·d^−1^ of P during the experiment. This constitutes around a third of the nutrients removed. Multiple studies have previously focused on the restoration ability of phytoplankton. For example, [39] observed that Pseudochlorella pringsheimii was able to remove 2.3 mg·L^−1^·d^−1^ of N and 1.3 mg·L^−1^·d^−1^ of P from aquaculture wastewater containing 34.8 and 18.6 mg·L^−1^ of N and P respectively. Ref. [40] grew Chlorella sp. in centrate wastewater containing between 150 and 340 mg·L^−1^ of N and between 90 and 300 mg·L^−1^ of P. The latter authors observed uptake rates between 14 and 20 mg·L^−1^·d^−1^ for N and of 2.8 mg·L^−1^·d^−1^ for P. In the present study, TN varied between 2.97 and 4.3 mg·L^−1^, while TP varied between 0.43 and 0.72 mg·L^−1^. The low nutrient concentrations will therefore have contributed to the reduced nutrient uptake rates. Furthermore, it is likely that the duckweed mat prevented light from entering the water column, inhibiting the phytoplankton growth and leading to a reduced nutrient uptake by this compartment.

The analysis of the biofilm revealed that this compartment released 11.2 mg of N and 4.9 mg of P (Table 3). The negative balance is probably associated with the reduction of the organic matter in the biofilm throughout the experiment.

The remaining third of nutrients removed from the water is probably linked to different processes such as sedimentation [41], denitrification and NH3-volatilisation [42]. Observed that, in diluted swine wastewater treated with the duckweed Spirodela oligorrhiza, 30% of TN present in the water is removed through ammonia volatilization [43]. Ammonia volatilization increases at pH values higher than 7 and a T close to 20 °C [44], two conditions consistent with the experimental conditions observed. Thus, it is possible that under the used experimental conditions part of the dissolved nitrogen was lost to the atmosphere.

Capturing the plant nutrients N and P using duckweed and using the plant biomass as part of a composite feed, can close the nutrient cycle and diminish the need for raw resources including mineable phosphate. However, the current experiment shows that under realistic conditions duckweed colonies capture just one-third of N and P removed from the medium. Thus, there is ample scope to improve the nutrient retention efficiency of duckweed systems, particularly by impeding phytoplankton growth and/or microbial activities.

## 5. Conclusions

The present study assessed the amounts of nutrients effectively recovered by duckweed from aquaculture effluents in a realistic, outdoor recirculating system. Phytoplankton in the water column removes a considerable amount of nutrients from the water. The extent of phytoplankton-mediated nutrient removal is likely to be particularly important in IMTA systems in seasons when duckweed cover is not present. However, in the long term, phytoplankton does not contribute to remediation as it cannot be easily removed from the water; hence, the nutrients will eventually be re-released into the water, nullifying the remediation process. In comparison, duckweed can be harvested, and if this is well managed, it can substantially contribute to the sequestration and removal of nutrients from aquaculture wastewater. This study confirms the phytoremediation ability of duckweed in aquaculture effluents; however, it also clearly shows that there is scope to further improve duckweed-based nutrient removal by impeding competing processes of algal growth and microbial activity.

## Figures and Tables

**Figure 1 plants-11-03103-f001:**
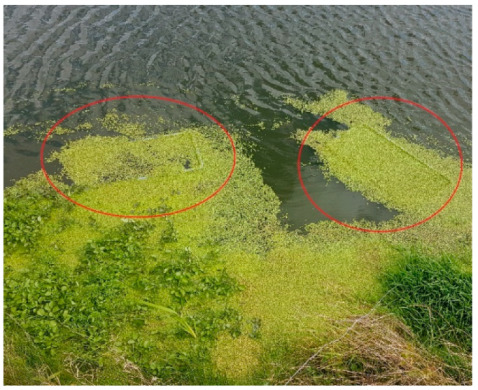
Tanks (circled in red) were submerged for a week in one of the duckweed-covered canals prior to being used for the mesocosm experiment. The pre-treatment resulted in the establishment of a biofilm on the internal surface of the tanks.

**Figure 2 plants-11-03103-f002:**
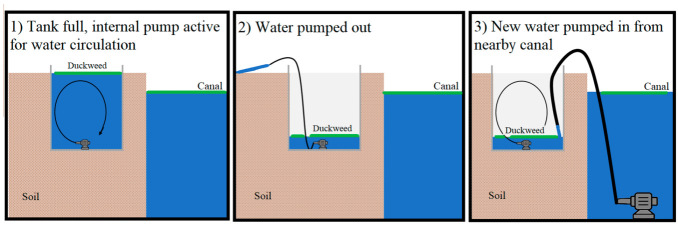
Experimental design. Tanks are buried next to a canal with aquaculture effluent. The tank is filled with aquaculture effluent (**1**). Twice a day, the tank is emptied without disrupting the biofilm or duckweed mat (**2**) and re-filled with fresh aquaculture effluent from the nearby canal (**3**).

**Figure 3 plants-11-03103-f003:**
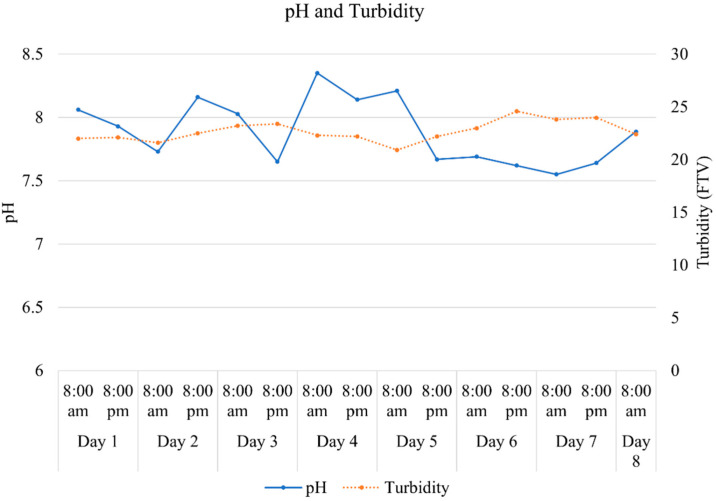
pH and turbidity in the aquaculture wastewater measured at 8:00 am and 8:00 pm during the eight days of the experiment. FTV = Formazin Turbidity Unit.

**Figure 4 plants-11-03103-f004:**
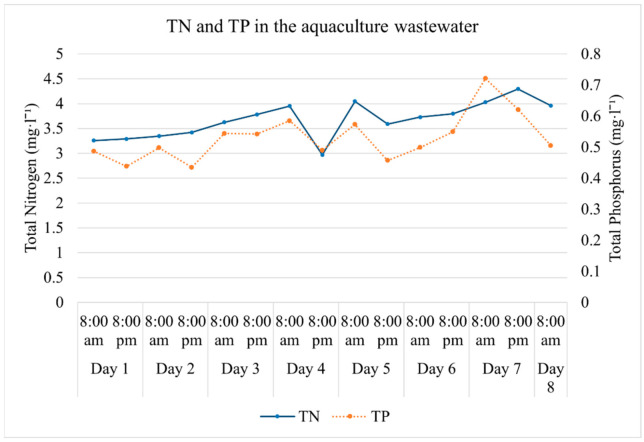
Total Nitrogen and Total Phosphorus concentrations in the aquaculture effluent measured at 8:00 am and 8:00 pm over the eight days of the experiment.

**Figure 5 plants-11-03103-f005:**
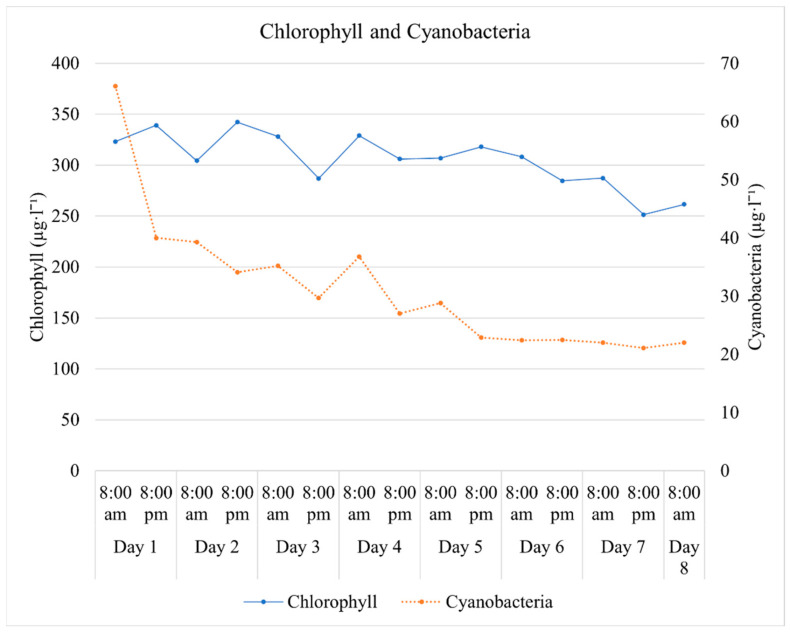
Algal chlorophyll and cyanobacteria concentration in the aquaculture wastewater measured at 8:00 am and 8:00 pm during the eight days of the experiment. N= night, D= day.

**Figure 6 plants-11-03103-f006:**
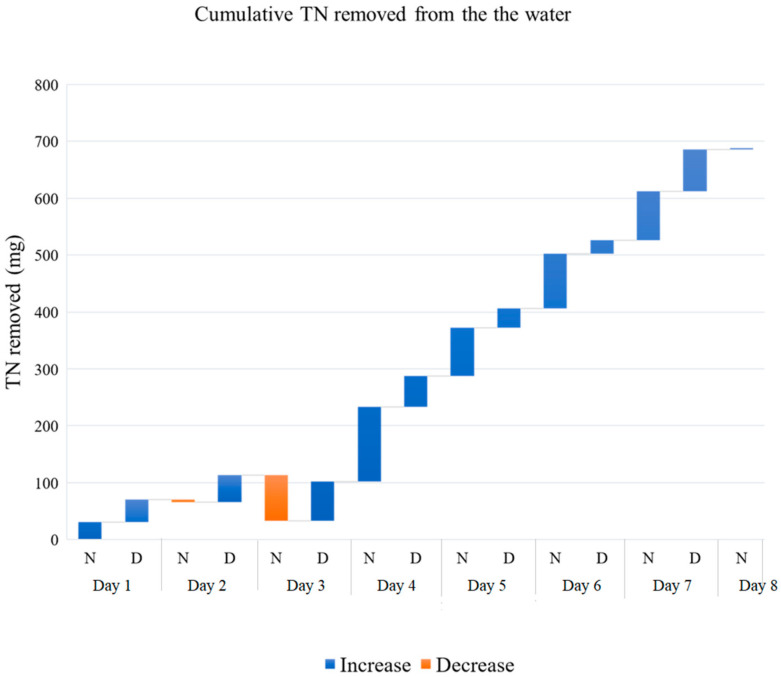
Cumulative Total Nitrogen removed from 70 L of aquaculture effluent every 12 h. N (Night) indicates the Total Nitrogen removed between 8:00 pm and 8:00 am; D (Day) indicates the Total Nitrogen removed between 8:00 am and 8:00 pm. The graph shows the average of three replicates.

**Figure 7 plants-11-03103-f007:**
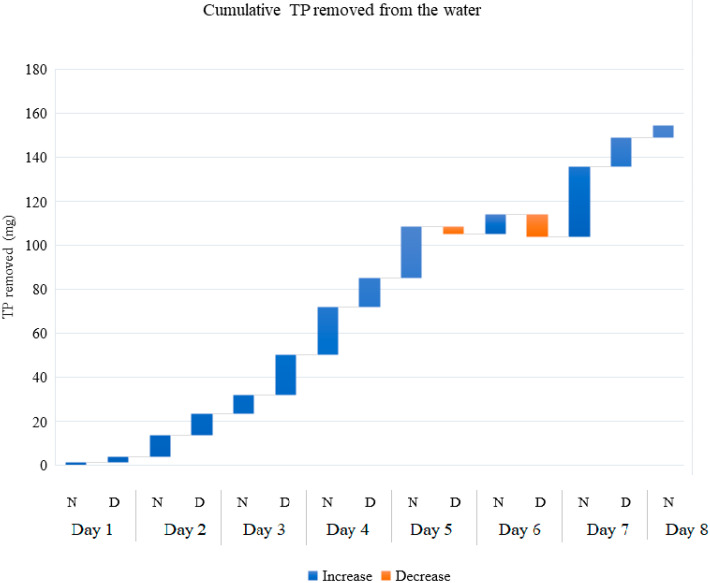
Cumulative Total Phosphorus removed from 70 L of aquaculture effluent every 12 h. N (Night) indicates the Total Phosphorus removed between 8:00 pm and 8:00 am; D (Day) indicates the Total Phosphorus removed between 8:00 am and 8:00 pm. The graph shows the average of three replicates.

**Figure 8 plants-11-03103-f008:**
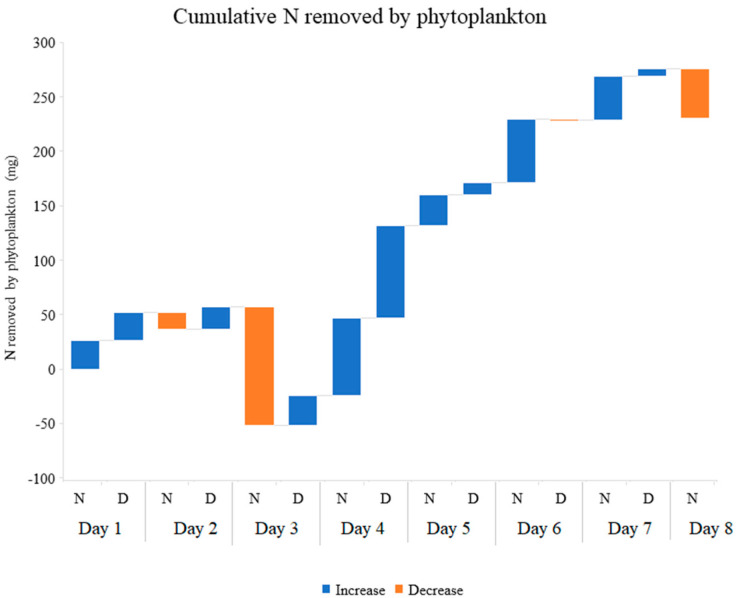
Cumulative Total Nitrogen removed by phytoplankton from 70 L of aquaculture effluent every 12 h. N (Night) indicates the Total Nitrogen removed between 8:00 pm and 8:00 am; D (Day) indicates the Total Nitrogen removed between 8:00 am and 8:00 pm. The graph shows the average of three replicates.

**Figure 9 plants-11-03103-f009:**
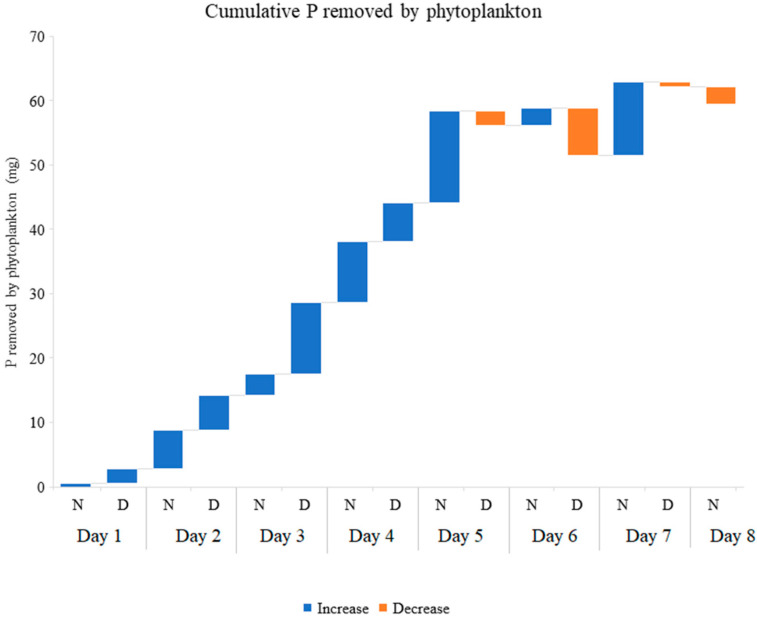
Cumulative total phosphorus removed from 70 L of aquaculture effluent every 12 h. N (Night) indicates the Total Phosphorus removed between 8:00 pm and 8:00 am; D (Day) indicates the Total Phosphorus removed between 8:00 am and 8:00 pm. The graph shows the average of three replicates.

**Figure 10 plants-11-03103-f010:**
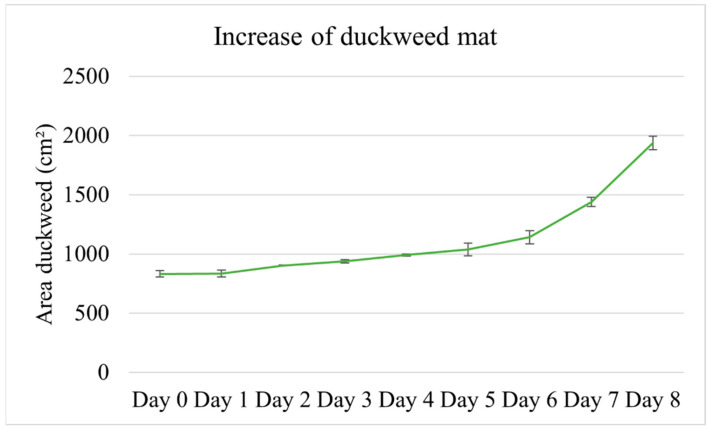
Increase in surface area covered by duckweed mat during the eight days of the experiment. Error bars are standard deviations.

**Figure 11 plants-11-03103-f011:**
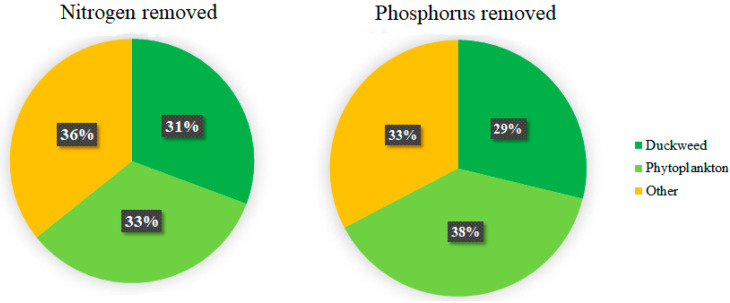
Percentages of total nitrogen and total phosphorus removed by the different compartments during the experiment.

**Table 1 plants-11-03103-t001:** Genera of microalgae identified in samples of biofilm and phytoplankton built in the internal surface of the tanks.

Genera Identified in the Phytoplankton and in the Biofilm
Bacillariophyta
*Cyclotella* sp.
*Tabellaria* sp.
*Nitzschia* sp.
Chlorophyta
*Micractinium* sp.
*Scenedesmus* sp.
*Actinastrum* sp.
*Pediastrum* sp.
*Chlamydomonas* sp.
*Monoraphidium* sp.
Charophyta
*Closterium* sp.

**Table 2 plants-11-03103-t002:** Biomass gained and nutrients accumulated by phytoplankton during the experiment. The total biomass produced was estimated from the phytoplankton filtered from the water samples collected twice a day from the inlet and outlet water.

Phytoplankton Biomass	
Total biomass that entered the system (g of fresh weight)	110.11
Total final biomass	142.75
N in tot. initial biomass (mg)	775.83
N in tot. final biomass (mg)	1005.89
**N accumulated by phytoplankton (mg)**	**230.06**
P in tot. initial biomass (mg)	199.3
P in tot. final biomass (mg)	258.38
**P accumulated by phytoplankton (mg)**	**59.08**

**Table 3 plants-11-03103-t003:** Biomass gained and nutrients removed by duckweed during the experiment.

Duckweed Biomass	
Initial dry weight (g)	3.32 ± 0.41
Final dry weight (g)	9.13 ± 0.55
N in initial biomass (mg·g^−1^)	38.4 ± 0.36
N in final biomass (mg·g^−1^)	36.8 ± 4.33
**N removed by duckweed (mg)**	**210.1 ± 71.3**
P in initial biomass (mg·g^−1^)	8.61 ± 0.49
P in final biomass (mg·g^−1^)	7.99 ± 0.37
**P removed by duckweed (mg)**	**44.5 ± 6.2**

**Table 4 plants-11-03103-t004:** Biofilm weight, % of organic matter, and nutrient content at the start and at the end of the experiment.

Biofilm	
Initial dry weight (g)	4.57 ± 3.51
Organic fraction (%)	15.63
Final dry weight (g)	6.02 ± 0.97
Organic fraction (%)	3.39
N in initial biofilm (mg·g^−1^)	5.0 ± 1.6
N in final biofilm (mg·g^−1^)	1. 7 ± 0.6
**N removed by biofilm (mg)**	**−11.2 ± 13.2**
P in initial biofilm (mg·g^−1^)	2.0 ± 0.4
P in final biofilm (mg·g^−1^)	0.5 ± 0.1
**P removed by biofilm (mg)**	**−4.9 ± 3.7**

## Data Availability

The data presented in this study are available on request from the corresponding author.

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
