# Peer review of "Integrated Multitrophic Aquaculture; Analysing Contributions of Different Biological Compartments to Nutrient Removal in a Duckweed-Based Water Remediation System"

_plants, 2022, doi:10.3390/plants11223103_

Round 1
Reviewer 1 Report
plants-2009974
Note: I am puzzled as to how I got this manuscript to review, since I have never published academically on duckweed nor on water remediation. I am not up to date on the state of the art in this field, beyond casual reading.
However, for the past 8 years I ran a duck-weed based water remediation/composting operation as a small scale hobby. So, combining my professional expertise and my hobby I hope I can provide a useful review of the results and presentation, although not of the literature coverage.
Abstract:
Good.
Figures & Tables (read firstly)
Figures 3 & 4 & 5
I would be happier if these figures had true numerical X axis of time, with perhaps shaded bands to indicate day/night.
The current figure is a 'line' graph with lines connecting sequential 'categorical' values on the X axis.
But, since measures are ~~evenly spaced~~ current plots cancel out to a visually similar presentation.
Figs 6-9
Y axis units incomplete or confusing.
Some legends specify 'from 70 l of water' but it is not clear if this information applies to all figures.
In any case, think through a more generalizable expression of Y axis units.
I appreciate the xx m N L-1 is not necessarily appropriate, because the water is flowing through the system.
But, a differently sized system would remove more (or less) elemental nutrient, so the units should attempt to be scale-independent.
Figure 9:
I am not sure about 'N removed by phytoplankton' etc.
Rather: 'N accumulated by phytoplankton...' because part of the point of the paper is that N and P entering phytoplankton is ultimately recycled within the system rather than removed through harvest.
Again, what is the total volume of water treated by the system?
These absolute removals do not mean much without the context of the water volume treated.
Discussion:
"The latter authors reviewed the char- 353
acteristics of freshwater aquaculture effluents and found that TN and TP in wastewater 354
generated by rainbow trout and perch farms ranges between 0.5 and 70 mg·l-1 and 0.42 355
and 15 mg·l-1 respectively, depending mainly on the fish density"
Too many layers of nesting; 'respectively' is almost always a bad idea.
I cannot tell whether the respectively refers to TN & TP, or trout vs. perch.-.
I note that studies cited in the Discussion have more generalizable units of mg N L-1 d-1 etc.
Materials & Methods:
Tanks were 35 x 60 x 45 cm but it is not clear which dimension is the depth.
Since duckweed only grows at the surface, a shallow tank would be expected to achieve better nutrient removal?
Question on design:
Waiting 8 days before duckweed harvest seems long.
In my amateur reactor (~7 x 12 m x 2 m depth) I scooped out ~60% of the duckweed coverage more than weekly without allowing it to achieve confluent growth; did biomass accumulation limit duckweed growth?
Oddly, there is no plot of the growth of duckweed coverage.
Some of the nutrient uptake plots suggest a slowing of growth
Results 3.4
Something is wrong.
I estimate the surface area of the tank at ~1575 cm^2, but according to Section 3.4 the duckweed accumulated to 19360 cm^x.
So the accumulated duckweed accumulated to 10X the area of a tank.
And the RGR is estimated as simply {ln(final) - ln(initial)}/time assuming continuous exponential accumulation over the entire time period.
See comments above; I see hints that growth actually slowed/stopped by about day 5.
Author Response
Dear reviewer, thank you very much for taking the time to review this work and for providing comments that improved our manuscript. Please, see below how we addressed all the points that you highlighted in your review:
Figures 3 & 4 & 5
I would be happier if these figures had true numerical X axis of time, with perhaps shaded bands to indicate day/night.
The current figure is a 'line' graph with lines connecting sequential 'categorical' values on the X axis. But, since measures are ~~evenly spaced~~ current plots cancel out to a visually similar presentation.
N and D in the figures were replaced with 8:00 am and 8:00 pm
Figs 6-9
Y axis units incomplete or confusing.
Some legends specify 'from 70 l of water' but it is not clear if this information applies to all figures.
We agree that the different text used for the captions generated confusion. All the captions now specify that the N and P removed refer to nutrients removed from 70 L every 12 hours
In any case, think through a more generalizable expression of Y axis units.
I appreciate the xx m N L-1 is not necessarily appropriate, because the water is flowing through the system. But, a differently sized system would remove more (or less) elemental nutrient, so the units should attempt to be scale-independent.
The aim of this study was to identify the fraction of nutrients used by duckweed and other compartments. For this reason, we need to show in the results TN and TP removed from 70 L every 12 hours (which is what we analysed in this study). However, we agree that scale-independent values are very important for readers that would like to compare their results with the results of this study. For this reason, we now calculated and included in the text the removal rates expressed in mg ·l-1·d-1:
“The removal of N rate varied between -0.16 and 2.63 mg ·l-1·d-1, while the removal rate of P varied between 0.05 and 0.64 mg ·l-1·d-1.” (Lines 354-355)
Figure 9:
I am not sure about 'N removed by phytoplankton' etc.
Rather: 'N accumulated by phytoplankton...' because part of the point of the paper is that N and P entering phytoplankton is ultimately recycled within the system rather than removed through harvest.
Agree, “removed” has been replaced by “accumulated”
Again, what is the total volume of water treated by the system?
These absolute removals do not mean much without the context of the water volume treated.
The caption for figure 9 now specify that the mg of N and P are removed from 70 L of effluent every 12 hours.
Discussion:
"The latter authors reviewed the characteristics of freshwater aquaculture effluents and found that TN and TP in wastewater generated by rainbow trout and perch farms ranges between 0.5 and 70 mg·l-1 and 0.42 and 15 mg·l-1 respectively, depending mainly on the fish density"
Too many layers of nesting; 'respectively' is almost always a bad idea.
I cannot tell whether the respectively refers to TN & TP, or trout vs. perch.-.
Thanks for the suggestion, the sentence has been rephrased: “The latter authors reviewed the characteristics of wastewater generated by rainbow trout and perch farms. They found that TN ranges between 0.5 and 70 mg·l-1, while TP ranges between 0.42 and 15 mg·l-1, depending mainly on the fish density.”
I note that studies cited in the Discussion have more generalizable units of mg N L-1 d-1 etc.
As specified above, we now included a more generalizable unit (rate expressed in mg/l/day) in lines 354-355
Materials & Methods:
Tanks were 35 x 60 x 45 cm but it is not clear which dimension is the depth.
The depth is now specified in the text: (Area 35x60 cm, depth 45cm)
Since duckweed only grows at the surface, a shallow tank would be expected to achieve better nutrient removal?
The real system next to our mesocosm has canals for the duckweed-based water remediation with a depth between 90 and 120 cm, so we also went for a system not too shallow. As showed in figure 2, water recirculation inside the tanks was provided by a pump. In this way the nutrients are constantly redistributed in the water column, and they stay available to duckweed.
Question on design:
Waiting 8 days before duckweed harvest seems long.
In my amateur reactor (~7 x 12 m x 2 m depth) I scooped out ~60% of the duckweed coverage more than weekly without allowing it to achieve confluent growth; did biomass accumulation limit duckweed growth?
We did not decide in advance the duration of the experiment. We decided to stop it when the surface was nearly completely covered by duckweed. We now better specified this in lines 195-196.
Oddly, there is no plot of the growth of duckweed coverage.
Some of the nutrient uptake plots suggest a slowing of growth
A figure (figure 10) showing the increase in duckweed mat has now been included. The growth rate increased the last two days. The graphs of the daily removal do not reflect exactly the growth rate as some nutrients can be released as a consequence of biomass (duckweed and phytoplankton) decomposition.
Results 3.4
Something is wrong.
I estimate the surface area of the tank at ~1575 cm^2, but according to Section 3.4 the duckweed accumulated to 19360 cm^x. So the accumulated duckweed accumulated to 10X the area of a tank.
The area of the tanks is 35x60=2100 cm2. However, there was indeed a typo. The final area 1936 cm2 the extra zero has been removed.
And the RGR is estimated as simply {ln(final) – ln(initial)}/time assuming continuous exponential accumulation over the entire time period.
See comments above; I see hints that growth actually slowed/stopped by about day 5.
RGR was calculated using Connolly & Wayne (1996): ln(final biomass/initial biomass)/days
Reviewer 2 Report
The manuscript entitled “Integrated Multitrophic Aquaculture; analysing contributions of different biological compartments to nutrient removal in a duckweed-based water remediation system” is the result of studies of the duckweed and microalgae influence in a reservoir under conditions close to natural. The conducted research is interesting in its approach and clearly described experiment. However, it requires some refinement in the description of methods and results.
The main conceptual comment:
The materials and methods do not adequately describe the “biological fractions” under study. Thus, phytoplankton, duckweed and biofilm are mentioned, but the descriptions of biofilm and phytoplankton in the results are not uniform.
Table 1 shows the microalgae taxonomic composition, defined to the genus, for phytoplankton such data are not given and no explanation is given for this.
Moreover, a new fraction appears in Figure 5 - cyanobacteria. How the authors isolated this fraction from phytoplankton (?) is unclear. If cyanobacteria predominated in phytoplankton, this should be shown at least in a table.
It would be logical to divide the genera of microalgae (for example, in Table 1) into larger taxa (Bacillariophyta, Chlorophyta, etc.).
Main comments:
The materials and methods do not provide brief descriptions of some measurement techniques:
Line 151 “algal chlorophyll content, cyanobacteria and turbidity” a brief description of the methodology is required, with reference to previously described applications.
Line 153 “Temperature and pH were also monitored” What instruments were used to measure?
Line 249-250 “The pH of the aquaculture effluent varied between 7.55 and 8.53 during the eight day 249 experiment, while the turbidity varied between 20.9 and 24.6 FTU (Fig. 3).” What about temperature? Why did the authors carry out these measurements?
Figure 10 and the text to which it refers - 36 and 33% is a significant proportion of nitrogen and phosphorus, almost one-third of the results and “other means” (Line 338) is a bit insufficient to describe. Add a more detailed description- what is meant.
Line 346 “these three types of organism” – the authors consider the constituent biota parts in the reservoir, the use of the term “types of organism” is inappropriate and misleading here. To my mind the most correct option is “three groups of organisms”.
Line 374-375 “However, the turbidity can explain the low number of taxa observed in the samples (Shen et al., 2011).” Do you mean Shen et al., 2011 or this study? In the present study, the taxonomic composition is only for the biofilm.
Specific comments:
Line 159 “Table 90.” – it's not clear what that means.
Line 175 add a description of the microscope - manufacturer and model.

Author Response
Dear reviewer, thank you very much for taking the time to review this work and for providing comments that improved our manuscript. Please, see below how we addressed all the points that you highlighted in your review:
The materials and methods do not adequately describe the “biological fractions” under study. Thus, phytoplankton, duckweed and biofilm are mentioned, but the descriptions of biofilm and phytoplankton in the results are not uniform.
A definition is now provided in lines 210-215
Table 1 shows the microalgae taxonomic composition, defined to the genus, for phytoplankton such data are not given and no explanation is given for this.
Microalgae in both biofilm and phytoplankton were observed. This is now specified in the table and caption
Moreover, a new fraction appears in Figure 5 - cyanobacteria. How the authors isolated this fraction from phytoplankton (?) is unclear. If cyanobacteria predominated in phytoplankton, this should be shown at least in a table.
Chlorophyll and cyanobacterial content in the water and water turbidity were measured using an AlgaeTorch produced by bbe-moldaenke GmbH.This info is provided in lines 184-187
We did not specifically assessed the amount of nutrients removed by cyanobacteria, we assessed nutrients removed by photosynthetic microalgae, which includes cyanobacteria. This is now specified in the definition of the biological compartments.
It would be logical to divide the genera of microalgae (for example, in Table 1) into larger taxa (Bacillariophyta, Chlorophyta, etc.).
The large taxa have been included in the table
Main comments:
The materials and methods do not provide brief descriptions of some measurement techniques:
Line 151 “algal chlorophyll content, cyanobacteria and turbidity” a brief description of the methodology is required, with reference to previously described applications.
This information is provided in lines 184-187
Line 153 “Temperature and pH were also monitored” What instruments were used to measure?
Added in line 187
Line 249-250 “The pH of the aquaculture effluent varied between 7.55 and 8.53 during the eight day 249 experiment, while the turbidity varied between 20.9 and 24.6 FTU (Fig. 3).” What about temperature? Why did the authors carry out these measurements?
As specified before, the aim of these measurements was to characterize the effluent. These results are discussed in section 4.1.
Figure 10 and the text to which it refers – 36 and 33% is a significant proportion of nitrogen and phosphorus, almost one-third of the results and “other means” (Line 338) is a bit insufficient to describe. Add a more detailed description- what is meant.
The sentence “such as denitrification, volatilization and sedimentation” was added. These mechanisms were not the object of the study and were not analysed specifically, but they are discussed in section 4 of the paper.
Line 346 “these three types of organism” – the authors consider the constituent biota parts in the reservoir, the use of the term “types of organism” is inappropriate and misleading here. To my mind the most correct option is “three groups of organisms”.
Replaced with “compartments”.
Line 374-375 “However, the turbidity can explain the low number of taxa observed in the samples (Shen et al., 2011).” Do you mean Shen et al., 2011 or this study? In the present study, the taxonomic composition is only for the biofilm.
The text reported is in lines are 476 and 477. “samples is now replaced by “biofilm and phytoplankton”
Specific comments:
Line 159 “Table 90.” – it's not clear what that means.
I can’t understand this comment. There is no “Table 90” in line 159
Line 175 add a description of the microscope - manufacturer and model.
Added
Round 2
Reviewer 2 Report
I am satisfied with the answers of the authors, thanks for the work done.